# Improving RENet by Introducing Modified Cross Attention for Few-Shot Classification

## Abstract

Few-shot classification is challenging since the goal is to classify unlabeled samples with very few labeled samples provided. It has been shown that cross attention helps generate more discriminative features for few-shot learning. This paper extends the idea and proposes two cross attention modules, namely the *cross scaled attention* (CSA) and the *cross aligned attention* (CAA). Specifically, CSA scales different feature maps to make them better matched, and CAA adopts the principal component analysis to further align features from different images. Experiments showed that both CSA and CAA achieve consistent improvements over state-of-the-art methods on four widely used few-shot classification benchmark datasets, *miniImageNet*, *tieredImageNet*, *CIFAR-FS*, and *CUB-200-2011*, while CSA is slightly faster and CAA achieves higher accuracies.

## 1 Introduction

Few-shot classification has drawn lots of attentions in recent years [52]. It originates from the observation that humans can learn new concepts with very few samples, and the goal is to classify unseen query samples given very few support samples. One may consider fine-tuning a pre-trained model using the labeled samples from the unseen classes; however, this usually causes severe overfitting, which can be alleviated by regularization and data augmentation but cannot be fully solved.

The meta-learning [17] has been widely used for few-shot learning recently. However, they usually do not focus enough on relevant features as shown in Figure 1, taking the prototypical network [46] for an example, and those irrelevant features causes the limitation of generalization to the unseen classes. The cross attention network (CAN) [18] and the relational embedding network (RENet) [20] remedy the above issue by proposing the cross attention. It has been shown that humans tend to locate the most relevant regions in the pair of labeled and unlabeled samples first to recognize a sample from an unseen class given a few labeled samples [18]. Inspired by that, CAN and RENet generate the attention maps across the support class features and the query sample features to make the network attends more on the target object regions.

In this work, we make improvements for RENet by further enhancing the feature discriminability for few-shot classification. We propose the *cross scaled attention* (CSA) and the *cross aligned attention* (CAA). CSA scales different feature maps to make them better matched. CAA further considers the alignment issue between different images by adopting the principal component analysis (PCA).

Our main contributions are as follows:

- We propose two cross attention modules, CSA and CAA, to improve RENet.
- Both proposed modules surpass the results of state-of-the-art methods on miniImageNet, tieredImageNet, CIFAR-FS, and CUB-200-2011.

Submitted to 36th Conference on Neural Information Processing Systems (NeurIPS 2022). Do not distribute.

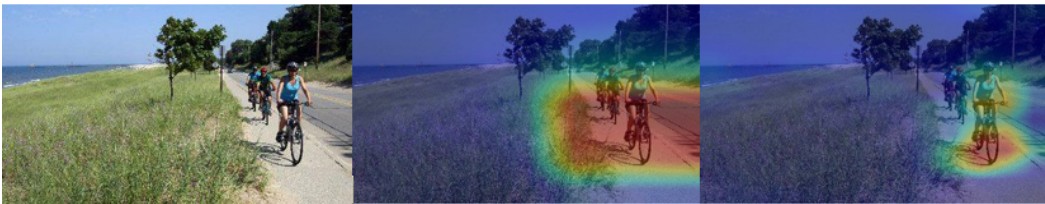

Figure 1. An example of the class activation maps [59] of an image (left) of an existing method [46] (middle) and our method (right). The warmer color indicates the higher value.

• CSA is slightly faster than CAA, while CAA achieves higher accuracies than CSA. Users can choose the one suitable for their needs.

The remaining of this paper is organized as follows. Section 2 provides background knowledge highly related to this work. Section 3 presents our approaches. Section 4 shows the experiment results. Section 5 concludes this work.

## 2   Related Work

**Few-Shot Classification**   Few-shot classification can be categorized into three groups, optimization-based methods [1, 29, 10, 44, 47], parameter-generating-based methods [3, 5, 32, 33], and metric-based methods [46, 48, 50, 18, 58, 20]. Optimization-based methods learn to update model parameters by designing the meta-learner as an optimizer. To adapt to new tasks efficiently for the learner, it learn a good initialization. Parameter-generating-based methods predict parameters by designing the meta-learner as a network. Metric-based methods learn an embedding function that maps images to a metric space such that the relevance between images is distinguished based on a distance metric.

Our method belongs to metric-based methods. The prototypical network [46], CAN [18], and RENet [20] are highly related to our work. Following CAN and RENet, we exploit the relation between the support set and query set. However, the prototypical network extracts the support and the query features independently which makes the model distracted by irrelevant features. The cross attention network improves the performance by using an attention network to refine features, which makes the model focus on the relevant regions. RENet further improves the performance by integrating a module that matchs the features in an image itself. Inspired by these works, we follow some of their structures and integrate a module that matchs the features between the support and the query images.

**RENet**   We follow the structure of RENet [20] and integrate our module to RENet. the *self-correlational representation* (SCR) and the *cross-correlational attention* (CCA) are proposed in RENet. SCR exploits the sliding window and the dilation to match the features in an image itself. CCA computes the cosine similarity between the support and the query images and generate attention maps. We consider the cross attention between the support and the query images by exploiting the sliding window and the dilation, which is similar to SCR. In addition to matching the features between the support and the query images, we also deal with the scaling and the alignment issues.

## 3   Approach

The network that addresses the challenge of generalization to unseen target classes is presented in this section. The overall structure is composed of five modules: an embedding module, SCR, CCA, and CSA/CAA, and a classification module. The embedding module extracts features of the input image. It consists of several cascaded convolutional layers, mapping an input image into a feature map. We use the ResNet-12 [16] network as our embedding module, which is identical to CAN [18] and RENet [20]. Following the prototypical network [46], CAN, and RENet, the support feature of a class is defined as the mean of its support set in the embedding space. The embedding module takes the support set and a query sample as inputs and produces the support feature map $Z_s$ and a query feature map $Z_q$. Each pair of feature maps ($Z_s$ and $Z_q$) are then fed through SCR, CCA, and CSA/CAA, which highlight the relevant regions and output more discriminative feature pairs ($s$ and

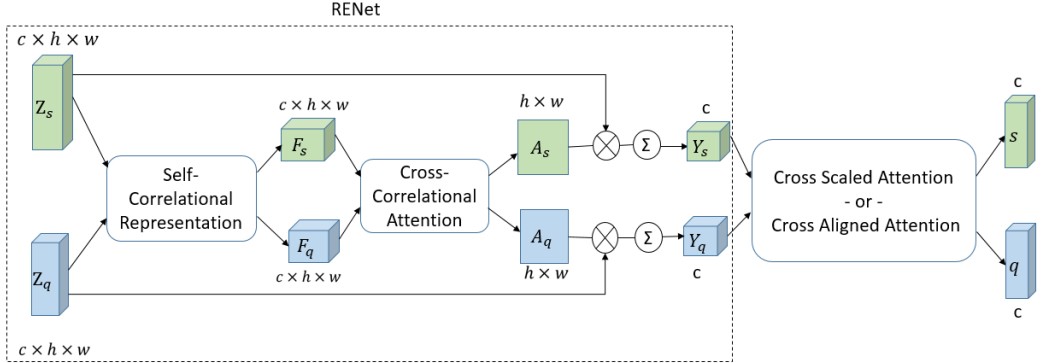

Figure 2: The overall architecture.

$q$) for classification. We first present a brief definition of the problem and a concise overview of the proposed architecture in Section 3.1 and Section 3.2 respectively. We then present technical details of CSA and CAA in Section 3.3 and describe our training objective in Section 3.4.

## 3.1 Problem Definition

The datasets for few-shot classification are split into the training set and the testing set, and each of them are further split into the support set and the query set. The support set contains few labeled samples and the query set contains unlabeled samples. Given the support set, few-shot classification aims to correctly classify the query set. The problem is called $N$-way $K$-shot if the support set is composed of $N$ classes and $K$ labeled samples per class.

Because deep neural networks are vulnerable to overfitting with few labeled samples [20], most few-shot classification methods adopt a meta-learning framework with episodic training. Following them, we adopt the episodic training mechanism, which has been shown effective for few-shot learning [46, 50, 45, 40, 15, 22, 10, 31].

## 3.2 Architecture Overview

The overall architecture is illustrated in Figure 2. For each pair of support classes and query samples, we obtain proper feature representations. The network can model and exploit the semantic relevance between the support feature and query feature. Our approach is different from many previous methods which extract the support and the query features independently. We resort to metric learning in this work. To be helpful to the subsequent matching, we integrate attention to the features.

The support feature map $Z_s \in \mathbb{R}^{c \times h \times w}$ is extracted from the support samples and the query feature map $Z_q \in \mathbb{R}^{c \times h \times w}$ is extracted from the query sample, where $c$, $h$, and $w$ denote the number of channels, height, and width of the feature maps respectively. The network generates attention maps for the input pair, which is then used to weight the feature map to achieve more discriminative feature representation, and the final outputs are $s$ and $q$. The architecture in Figure 2 consists of three main learnable modules: SCR, CCA, and CSA or CAA. Since SCR and CCA have already proposed in RENet [20], we start our description from CSA and CAA. More detail can refer to [20].

## 3.3 Cross Scaled Attention (CSA) and Cross Aligned Attention (CAA)

Figure 3 illustrates the structure of CSA and CAA. Inspired by SCR [20], We propose two similar modules CSA and CAA. SCR only considers about the correlation in the image itself, and we further think about the correlation between the support and the query images. CCA [20] also consider the correlation between the support and the query images. It computes the cosine similarity between the support and the query images and generate attention maps. On the other hand, CSA and CAA match the features between the support and the query images by computing the Hadamard product. They further help our model focus on more important features. SCR focuses on the target object in an image, and CSA and CAA focus on the target objects in both the support and the query images. The

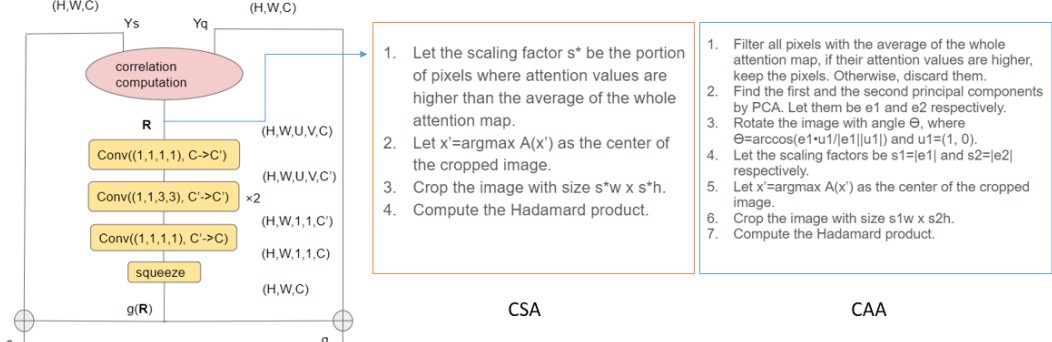

Figure 3: The structure of CSA and CAA.

structure is almost identical to SCR, and the only difference is the input. Similar to CCA, CSA and CAA take an input pair of support and query, $Y_s$ and $Y_q$, and produces the final embeddings, $s$ and $q$.

**Correlation computation**   Similar to SCR, we exploit the sliding window and the dilation to match the features. However, instead of matching the features of each position and its neighborhood which is presented in SCR, we match the features between the support and the query. The Hadamard product of a vector at each position $\mathbf{x} \in [1, H] \times [1, W]$ and vectors at the neighborhood of $\mathbf{x'} \in [1, H] \times [1, W]$ is computed and collected into a cross-correlation tensor $\mathbf{R}$. We represent the tensor $\mathbf{R}$ as a function with a vector output:

$$\mathbf{R}(\mathbf{x}, \mathbf{x'}, \mathbf{p}) = \frac{Y_s(\mathbf{x})}{\|Y_s(\mathbf{x})\|} \odot \frac{Y_q(\mathbf{x'} + \mathbf{p})}{\|Y_q(\mathbf{x'} + \mathbf{p})\|}, \tag{1}$$

where $\mathbf{p} \in [-d_U, d_U] \times [-d_V, d_V]$. It corresponds to a relative position in the neighborhood window such that $2d_U + 1 = U$ and $2d_V + 1 = V$, which includes the center position. The edges of the feature map are zero-padded for sampling off the edges.

To make the training process more efficient, we do not iterate through the whole image for $\mathbf{x'}$. We take the position at attention map produced previously that has maximum value as the center and crop the image to find the scope of the target object, where the region we iterate through. For CSA, the size of the cropped region is $s^*w \times s^*h$, where $w$ and $h$ are the width and the height of the image respectively, and $s^*$ is the scaling factor which is the portion of pixels where attention values are higher than the average of the whole attention map. If we change the scaling factor, which is the only difference between CSA and CAA, the module will become CAA. We elaborate on how we tune the scaling factor in the next two paragraphs.

**Cross scaled attention (CSA)**   In this paragraph, we first elaborate on how we tune the scaling factor. In general, we have two scaling factors, $s_1$ and $s_2$, to crop the image with size $s_1w \times s_2h$, where $w$ and $h$ are the width and the height of the image respectively. We tune the scaling factors $s_1$ and $s_2$ in three different ways. Firstly, we let $s^* = s_1 = s_2$ and fix $s^*$ to 0.5. Secondly, we let $s^*$ be the portion of pixels where attention values are higher than the average of the whole attention map, which is the scaling factor adopted in CSA. Finally, we adopt PCA to determine the scaling factors, and this is what CAA does. We find out that the first method achieves the lowest accuracy, so we regard it as a baseline. CAA achieves higher accuracies than CSA, but its training time is longer compared to CSA. The results of different methods are presented in Section 4.2.

The scaling factor $s^*$ for CSA is obtained by the following equation:

$$s^* = \frac{N}{A}, \tag{2}$$

where $N$ is the number of pixels with attention values higher than the average of the attention map, and $A$ is the is the total number of pixels.

**Cross aligned attention (CAA)**   We first filter all the pixels with the threshold of the average value of the whole attention map. If the attention values of the pixels are lower than the threshold, we

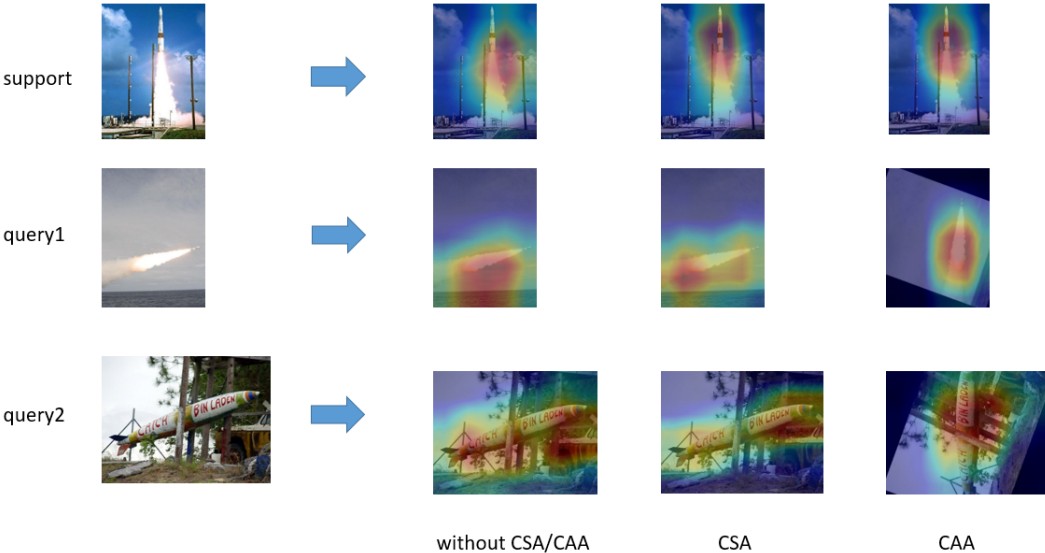

Figure 4. The class activation maps [59] of the support image (top) and the query images (middle and bottom). In CAA, the target objects in the support and the query images are aligned after rotation. The warmer color indicates the higher value.

will discard them. PCA is then conducted to find the first and the second principal components, and the image is rotated with the angle $\theta$, where $\theta$ is the angle between the first principal component and the horizontal line. We crop the image with size $s_1 w \times s_2 h$ centered at the position where the attention value is maximum, where $s_1$ and $s_2$ are the magnitudes of the first and the second principal components respectively, and $w$ and $h$ are the width and the height of the image respectively.

Consider a data matrix $X$ with column-wise zero empirical means, which indicates that the sample mean of each column has been shifted to zero. The transformed is defined by a set of coefficient vectors $v$, and each coefficient vector is constrained to be a unit vector. To maximize variance, the first coefficient vector $v_1$ has to satisfy the following equation:

$$v_1 = argmax(\frac{v^T X^T X v}{v^T v}). \tag{3}$$

With $v_1$ found, the first principal component is $e_1 = X v_1 v_1^T$.

The second principal component $e_2 = X v_2 v_2^T$ can be found by the second coefficient vector $v_2$. $v_2$ can be found by the following equations:

$$\hat{X} = X - X v_1 v_1^T, \tag{4}$$

$$v_2 = argmax(\frac{v^T \hat{X}^T \hat{X} v}{v^T v}). \tag{5}$$

The rotation angle $\theta$ can be derived from the following equation:

$$\theta = \cos^{-1}(\frac{e_1 \cdot u_1}{\|e_1\|\|u_1\|}), \tag{6}$$

where $u_1$ is the horizontal unit vector $(1, 0)$.

As shown in Figure 4, CAA aligns the target objects in the support and the query images since we rotate the image to help us match the features of the target objects in both images.

**Cross attention learning**    A series of 2D convolutions is applied to analyze the self-correlation patterns in $\mathbf{R}$. For computational efficiency, the convolutional block follows a bottleneck structure as shown in Figure 3. It consists of a point-wise convolution layer for channel size reduction, two $3 \times 3$ convolution layers for transformation, and another point-wise convolution layer for channel size

recovery. We insert batch normalization and ReLU between the convolutions. The spatial dimensions of local correlation patterns are reduced from $U \times V$ to $1 \times 1$ such that the output $g(\mathbf{R})$ and $Y_s$ ($Y_q$) has the same size since they are gradually aggregated by the convolution block $g(\cdot)$ without padding. The process of analyzing structure patterns could be complementary to appearance patterns in the representation $Y_s$ ($Y_q$). Therefore, we combine the two representations to produce the final embeddings $s$ and $q$:

$$s = g(\mathbf{R}) + Y_s, \tag{7}$$
$$q = g(\mathbf{R}) + Y_q, \tag{8}$$

which reinforces the base features with relational features and helps the few-shot learner focus on the target objects in the images.

## 3.4 Training and Testing (Inference)

**Training** Following [18] and [20], we train the network via minimizing the classification loss on the query samples of the training set. The classification module is composed of the nearest neighbor classifier and a global classifier. The nearest neighbor classifier classifies the query samples into N support classes based on pre-defined similarity measures. Each position in the query feature maps is constrained to be correctly classified to obtain precise attention maps. We define the nearest neighbor classification loss $L_1$ as the negative log-probability according to the true class label. A fully connected layer followed by *softmax* to classify each query sample among all available training classes is used in the global classifier. We compute the global classification loss $L_2$. Finally, we define the overall classification loss as $L = \lambda L_1 + L_2$, where $\lambda$ is the weight to balance the effects of different losses. We train the network end-to-end by optimizing $L$ with the stochastic gradient descent algorithm.

**Testing (Inference)** Many existing methods including the prototypical network [46] and RENet [20] use the inductive inference. The global average pooling is performed to the features to get the mean support and query features. The label for a query sample is predicted by finding the class which has the nearest mean support feature under a distance metric.

However, each class has very few labeled samples in few-shot classification task, so the support features of classes can hardly represent the true class distribution. To alleviate the problem, [18] proposed a simple and effective transductive inference algorithm that utilizes the unlabeled query samples to enrich the support features of classes.

Following [18], we use the transductive inference. In this way, the support features of classes can be more representative and robust. Experiment shows that the transductive inference achieves higher performance than the inductive inference especially in 1-shot where the problem described above is more serious.

# 4 Experiment Results

## 4.1 Experiment Setup

**Datasets** We use four standard benchmarks for few-shot classification for evaluation: miniImageNet, tieredImageNet, CIFAR-FS, and CUB-200-2011 (Caltech-UCSD Birds-200-2011).

- miniImageNet [50] is a subset of ImageNet (ILSVRC-2012) [21] which consists of 60000 images. It contains 100 object classes with 600 images per class. These classes are randomly split into 64, 16, and 20 classes for training, validation, and testing respectively. All images are of size $84 \times 84$.
- tieredImageNet [42] is a much larger subset of ImageNet (ILSVRC-2012) [21]. It contains 608 classes grouped into 34 high-level categories. These are divided into 20, 6, and 8 categories for training, validation, and testing respectively, which corresponds to 351, 97, and 160 classes for training, validation, and testing respectively. All images are of size $84 \times 84$.
- CIFAR-FS [2] is a subset of CIFAR-100 which consists of 60000 images. It contains 100 object classes with 600 images per class. These classes are randomly split into 64, 16, and 20 classes for training, validation, and testing respectively. All images are of size $32 \times 32$.

- CUB-200-2011 [53] is an image dataset with photos of 200 bird species (mostly North American). It consists of 100, 50, and 50 classes for training, validation, and testing respectively.

**Experiment setting**    We conduct experiments for our approach on 5-way 1-shot and 5-way 5-shot settings. For an $N$-way $K$-shot setting, we form the episode with $N$ classes and each class includes $K$ support samples. We use 15 query samples per class in an episode for both training and testing. We randomly sample 2000 episodes from the testing set when testing. The average accuracy and the corresponding 95% confidence interval are reported over the 2000 episodes.

**Implementation details**    We use Pytorch to implement all our experiments on one NVIDIA RTX-3080 GPU. The ResNet-12 [16] network is used as our embedding module. The input images size is $84 \times 84$ for miniImageNet and tieredImageNet, and $32 \times 32$ for CIFAR-FS. Horizontal flip, random crop, and random erasing are adopted as data augmentation during training. We use SGD as the optimizer. Each batch contains 8 episodes. For miniImageNet, CIFAR-FS, and CUB-200-2011, the model is trained for 90 epochs, with each epoch consisting of 1200 episodes, and the initial learning rate is 0.1 and decreased to 0.006, 0.0012, and 0.00024 at 60, 70, and 80 epochs, respectively. For tieredImageNet, the model is trained for 80 epochs, with each epoch consisting of 13980 episodes, and the initial learning rate is set to 0.1 with a decay factor of 0.1 at every 20 epochs. We set the temperature hyperparameter [20] to 2 for CUB-200-2011 and 5 otherwise, and we set the weight hyperparameter ($\lambda$) in the overall loss function to 0.25, 0.5, and 1.5 for ImageNet derivatives, CIFAR-FS, and CUB-200-2011 respectively. We set U=5 and V=5 in our experiment. We cross-validate all hyperparameters in the validation sets and fix them afterward in all experiments.

**Comparison with state-of-the-art methods**    Table 1 shows the comparison between our method and existing few-shot methods[1] on miniImageNet, tieredImageNet, CIFAR-FS, and CUB-200-2011. All results in Table 1 except our work are directly adopted from their papers. "-" indicates the results are not available in their papers. Many existing methods extract features of support and query samples independently, making the features focus on the non-target objects. To avoid the issue, CAN [18], RENet [20], and our method highlights the target object regions and gets more discriminative features instead. Compared to CAN and RENet, our method achieves higher accuracies.

## 4.2  Ablation Study

We show the effectiveness of each component of the network by empirical results and compare the time cost in this subsection. In [18] and [20], a series of experiments in their ablation study has already been completed. Following them, we experiment on miniImageNet in this subsection. We show the effectiveness of CAA and compare the performances of CAA and CSA. We firstly introduce a baseline to be used for comparison. If we remove SCR, CCA, and CSA/CAA, the model almost become the prototypical network [46] with ResNet-12 [16] as the backbone, and the only difference is a global classifier. Therefore, we create a variant named R12-proto by removing SCR, CCA, and CSA/CAA. In R12-proto, the features from the embedding module are directly fed to the nearest neighbor and global classifier, and the model is trained with the joint of global and nearest neighbor classification loss. The comparison between all variants are shown in Table 2. Time cost is shown in Table 3.

**Influence of SCR, CCA, and CSA/CAA**    By comparing RENet+CSA/CAA and R12-proto, we observe consistent improvements on both 1-shot and 5-shot scenarios as shown in Table 2. The reason is that when using SCR, CCA, and CSA/CAA, our model can highlight the relevant regions and extract more discriminative features. The performance gap shows that (1) conventionally independently extracted features tend to focus on the non-target regions and produce inaccurate similarities. (2) SCR, CCA, and CSA/CAA can help to highlight target regions and reduce such inaccuracy. As shown in Table 2, RENet+CSA/CAA outperforms R12-proto consistently, which further demonstrates the effectiveness of the attention mechanism.

**Influence of CSA and CAA**    To verify the effectiveness of CSA and CAA, we test another variant without the modules. We remove the component of CSA/CAA. That is, after we get the feature maps

---

[1]We re-implement the prototypical network with ResNet-12 as the backbone in Table 1.

Table 1: Performance comparison in terms of accuracy (%) with 95% confidence intervals on 5-way classification on (a) miniImageNet and tieredImageNet and (b) CIFAR-FS and CUB-200-2011.

(a) Results on miniImageNet and tieredImageNet datasets.

| Model | Backbone | miniImageNet | | tieredImageNet | |
|---|---|---|---|---|---|
| | | 1-shot | 5-shot | 1-shot | 5-shot |
| MAML [10] | ConvNet | 48.70 ± 0.84 | 55.31 ± 0.73 | 51.67 ± 1.81 | 70.30 ± 1.75 |
| *cosine* classifier [8] | ResNet-12 | 55.43 ± 0.81 | 77.18 ± 0.61 | 61.49 ± 0.91 | 82.37 ± 0.67 |
| MTL [47] | ResNet-12 | 61.20 ± 1.80 | 75.50 ± 0.80 | - | - |
| TADAM [36] | ResNet-12 | 58.50 ± 0.30 | 76.70 ± 0.30 | - | - |
| PPA [39] | WRN-28-10 | 59.60 ± 0.41 | 73.74 ± 0.19 | 65.65 ± 0.92 | 83.40 ± 0.65 |
| wDAE-GNN [15] | WRN-28-10 | 61.07 ± 0.15 | 76.75 ± 0.11 | 68.18 ± 0.16 | 83.09 ± 0.12 |
| SimpleShot [51] | ResNet-18 | 62.85 ± 0.20 | 80.02 ± 0.14 | - | - |
| TPN [28] | ResNet-12 | 59.46 | 75.65 | 59.91 ± 0.94 | 73.30 ± 0.75 |
| RFS-simple [49] | ResNet-12 | 62.02 ± 0.63 | 79.64 ± 0.44 | 69.74 ± 0.72 | 84.41 ± 0.55 |
| LEO [44] | WRN-28-10 | 61.76 ± 0.08 | 77.59 ± 0.12 | 66.33 ± 0.05 | 81.44 ± 0.09 |
| MetaOpt [22] | ResNet-12 | 62.64 ± 0.62 | 78.63 ± 0.46 | 65.99 ± 0.72 | 81.56 ± 0.53 |
| adaNet [33] | ResNet-12 | 56.88 ± 0.62 | 71.94 ± 0.57 | - | - |
| DC [26] | ResNet-18 | 62.53 ± 0.19 | 79.77 ± 0.19 | - | - |
| Shot-Free [41] | ResNet-12 | 59.04 | 77.64 | 63.52 | 82.59 |
| S2M2 [30] | ResNet-34 | 63.74 ± 0.18 | 79.45 ± 0.12 | - | - |
| MN [50] | ConvNet | 43.44 ± 0.77 | 60.60 ± 0.71 | - | - |
| MN [50] | ResNet-12 | 63.08 ± 0.80 | 75.99 ± 0.60 | 68.50 ± 0.92 | 80.60 ± 0.71 |
| RN [48] | ConvNet | 50.44 ± 0.82 | 65.32 ± 0.70 | 54.48 ± 0.93 | 71.32 ± 0.78 |
| PN [46] | ConvNet | 49.42 ± 0.78 | 68.20 ± 0.66 | 53.31 ± 0.89 | 72.69 ± 0.74 |
| PN [46] | ResNet-12 | 60.26 ± 0.49 | 73.65 ± 0.37 | 64.56 ± 0.56 | 76.78 ± 0.43 |
| NegMargin [27] | ResNet-12 | 63.85 ± 0.81 | 81.57 ± 0.56 | - | - |
| CTM [23] | ResNet-18 | 64.12 ± 0.82 | 80.51 ± 0.13 | 68.41 ± 0.39 | 84.28 ± 1.73 |
| FEAT [56] | ResNet-12 | 66.78 ± 0.20 | 82.05 ± 0.14 | 70.80 ± 0.23 | 84.79 ± 0.16 |
| DeepEMD [58] | ResNet-12 | 65.91 ± 0.82 | 82.41 ± 0.56 | 71.16 ± 0.87 | 86.03 ± 0.58 |
| CAN [18] | ResNet-12 | 63.85 ± 0.48 | 79.44 ± 0.34 | 69.89 ± 0.51 | 84.23 ± 0.37 |
| CAN+T [18] | ResNet-12 | 67.19 ± 0.55 | 80.64 ± 0.35 | 73.21 ± 0.58 | 84.93 ± 0.38 |
| RENet [20] | ResNet-12 | 67.60 ± 0.44 | 82.58 ± 0.30 | 71.61 ± 0.51 | 85.28 ± 0.35 |
| **RENet+CSA (ours)** | ResNet-12 | 73.18 ± 0.51 | 84.20 ± 0.31 | 75.58 ± 0.57 | 85.74 ± 0.39 |
| **RENet+CAA (ours)** | ResNet-12 | **73.61 ± 0.51** | **84.43 ± 0.30** | **76.71 ± 0.55** | **86.38 ± 0.35** |

(b) Results on CIFAR-FS and CUB-200-2011 datasets.

| Model | Backbone | CIFAR-FS | | CUB-200-2011 | |
|---|---|---|---|---|---|
| | | 1-shot | 5-shot | 1-shot | 5-shot |
| MAML [10] | ConvNet | 58.9 ± 1.9 | 71.5 ± 1.0 | - | - |
| MAML [10] | ResNet-34 | - | - | 67.28 ± 1.08 | 83.47 ± 0.59 |
| *cosine* classifier [8] | ResNet-12 | - | - | 67.30 ± 0.86 | 84.75 ± 0.60 |
| *cosine* classifier [8] | ResNet-34 | 60.39 ± 0.28 | 72.85 ± 0.65 | - | - |
| MetaOpt [22] | ResNet-12 | 72.6 ± 0.7 | 84.3 ± 0.5 | - | - |
| Shot-Free [41] | ResNet-12 | 69.2 | 84.7 | - | - |
| RFS-simple [49] | ResNet-12 | 71.5 ± 0.8 | 86.0 ± 0.5 | - | - |
| NegMargin [27] | ResNet-18 | - | - | 72.66 ± 0.85 | 89.40 ± 0.43 |
| S2M2 [30] | ResNet-34 | 62.77 ± 0.23 | 75.75 ± 0.13 | 72.92 ± 0.83 | 86.55 ± 0.61 |
| Boosting [13] | WRN-28-10 | 73.6 ± 0.3 | 86.0 ± 0.2 | - | - |
| FEAT [56] | ResNet-12 | - | - | 73.27 ± 0.22 | 85.77 ± 0.14 |
| MN [50] | ResNet-12 | - | - | 71.87 ± 0.85 | 85.08 ± 0.57 |
| RN [48] | ConvNet | 55.0 ± 1.0 | 69.3 ± 0.8 | - | - |
| RN [48] | ResNet-34 | - | - | 66.20 ± 0.99 | 82.30 ± 0.58 |
| PN [46] | ResNet-12 | 70.21 ± 0.52 | 80.60 ± 0.40 | 66.09 ± 0.92 | 82.50 ± 0.58 |
| DeepEMD [58] | ResNet-12 | - | - | 75.65 ± 0.83 | 88.69 ± 0.50 |
| CAN [18] | ResNet-12 | 71.65 ± 0.50 | 83.72 ± 0.38 | - | - |
| CAN+T [18] | ResNet-12 | 76.61 ± 0.56 | 84.37 ± 0.38 | - | - |
| RENet [20] | ResNet-12 | 74.51 ± 0.46 | 86.60 ± 0.32 | 79.49 ± 0.44 | 91.11 ± 0.24 |
| **RENet+CSA (ours)** | ResNet-12 | 80.02 ± 0.51 | 87.63 ± 0.33 | 85.89 ± 0.45 | 92.03 ± 0.25 |
| **RENet+CAA (ours)** | ResNet-12 | **80.40 ± 0.50** | **87.76 ± 0.33** | **86.63 ± 0.44** | **92.88 ± 0.22** |

Table 2: Ablation study on miniImageNet with performance comparison in terms of accuracy (%).

| Variant | 5-way 1-shot | 5-way 5-shot |
|---|---|---|
| R12-proto | 66.36 | 75.65 |
| without CSA/CAA | 72.81 | 83.84 |
| unscaled | 72.91 | 84.06 |
| **with CSA** | 73.18 | 84.20 |
| **with CAA** | **73.61** | **84.43** |

Table 3: Time cost on four datasets. All models are implemented in PyTorch and tested on Nvidia RTX-3080.

| Time | Model | miniImageNet | tieredImageNet | CIFAR-FS | CUB-200-2011 |
|---|---|---|---|---|---|
| training | RENet | 5 h 41 m | 48 h 05 m | 5 h 43 m | 1 h 08 m |
| | **RENet+CSA** | 5 h 52 m | 48 h 45 m | 5 h 55 m | 1 h 10 m |
| | **RENet+CAA** | 6 h 16 m | 49 h 17 m | 6 h 20 m | 1 h 15 m |
| inference | RENet | 2 m 06 s | 2 m 10 s | 2 m 07 s | 2 m 03 s |
| | **RENet+CSA** | 2 m 08 s | 2 m 11 s | 2 m 09 s | 2 m 04 s |
| | **RENet+CAA** | 2 m 12 s | 2 m 15 s | 2 m 14 s | 2 m 08 s |

$Y_s$ and $Y_q$ from CCA, the features are fed to the nearest neighbor and global classifier, and the model is trained with the joint of global and nearest neighbor classification loss. As shown in Table 2, both the network with CSA and the network with CAA outperform the variant model. The improvement indicates that CSA and CAA can help to highlight target regions more effectively compared to the model without CSA/CAA.

**Influence of PCA** Using PCA achieves higher accuracies for our model compared to other methods mentioned in Section 3.3. We compare the performances of the unscaled version, CSA, and CAA in Table 2. As described in Section 3.3, the unscaled version is the variant whose scaling factor is fixed to 0.5. As shown in Table 2, CAA achieves the highest accuracy, and we conclude that the alignment of the target objects in the support and the query images benefits classification.

**Speed comparison** We compare the training time and inference time of RENet, RENet+CSA, and RENet+CAA in Table 3. As can be seen, the training time of RENet+CSA is slightly longer than RENet, and the training time of RENet+CAA is slightly longer than RENet+CSA but not by much. The inference time of RENet+CSA is longer than RENet, and the training time of RENet+CAA is longer than RENet+CSA, but the differences are so slim that they are practically insignificant.

## 5 Conclusion

This work improves RENet for few-shot classification by introducing two cross attention modules, CSA and CAA, which model the semantic relevance between the support and the query features. Specifically, CSA scales different feature maps to make them better matched, and CAA adopts the principal component analysis to further align features from different images. As a result, the proposed modules focus on more relevant regions by considering both the support and the query images rather than only the latter ones. Empirically, RENet with both CSA and CAA outperformed state-of-the-art methods on miniImageNet, tieredImageNet, CIFAR-FS, and CUB-200-2011, four widely used datasets for few-shot learning, in terms of accuracy. The ablation study further verified that the improvements are achieved owing to the proposed modules.

Our work indicated that in few-show learning information contained in those few support samples should be exploited as much as possible, and the cross attention is one such way to do it. Although such techniques may require slightly longer training time, we believe that it is worthwhile especially in the scenarios where labeled data are valuable and few.

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
