Table 4: Results of the inductive inference on miniImageNet in terms of accuracy (%).

| Variant | 5-way 1-shot | 5-way 5-shot |
|---------|--------------|--------------|
| inductive | 69.41 | 83.30 |

# A   Appendix

**Influence of the transductive inference**   The transductive inference greatly improves our model compared to the inductive inference especially in 1-shot where the problem described in Section 3.4 is more serious. To be fair, we adopt the inductive inference with our model and compare with other methods. As shown in Table 1 and Table 4, our model with the inductive inference still outperforms other methods, although the margins between them are not that much compared to the transductive inference.