# OpenReview forum: "Improving RENet by Introducing Modified Cross Attention for Few-Shot Classification"
_NeurIPS.cc/2022/Conference — NeurIPS 2022 Submitted_

### Official Review · Reviewer_8rzp · 2022-07-04

**Rating:** 3
**Confidence:** 5
**Soundness:** 2 fair
**Presentation:** 2 fair
**Contribution:** 2 fair

**Summary:**

This paper aims to address the problem of few-shot learning. Inspired by the recent advances in cross attention, this paper explores to extend this paper and proposes two cross attention modules, i.e., cross scaled attention and cross aligned attention. Particularly, with the help of principal component analysis, cross aligned attention tries to align features from different images. In the experiments, the proposed method is evaluated on multiple datasets.

**Questions:**

I am not clear about the motivation of using principal component analysis.

**Limitations:**

The authors have addressed the limitations and potential negative societal impact of their work.

**Strengths And Weaknesses:**

Using cross attention to extract discriminative features is reasonable for few-shot learning task.

Weakness:

1. The motivation of this paper is not clear. The Introduction is very simple. I am not clear about the motivation of the proposed cross scaled attention and the cross aligned attention. Meanwhile, the authors should interpret the advantages of the proposed method. Finally, I am not clear why use principal component analysis. I recommend the authors revise this section carefully.

2. The proposed method only consists of some commonly used operations. And this paper does not propose any inspired ideas. Therefore, this paper lacks novelty. Meanwhile, during introducing approach details, the authors should further interpret the motivation of the used operations, which is helpful for better understanding of the proposed method.

3. The writing of the approach section is tedious. I advise the authors to modify the method section carefully. Meanwhile, I am not clear why the proposed method could address the problem of few samples. In the experiments, the authors do not sufficiently analyze the proposed method. The authors should give more visualization examples and training curves to show the effectiveness of the proposed method.

---

### Official Review · Reviewer_5kD8 · 2022-07-06

**Rating:** 3
**Confidence:** 4
**Soundness:** 2 fair
**Presentation:** 1 poor
**Contribution:** 2 fair

**Summary:**

The paper extends RENet[20] by adding two variants of attention modules that rotate and crop the query image around the predicted (by attention) target object that best matches the support sample via dense matching of the feature maps. Good results are demonstrated on mini/tiered-imagenet, cub, and cifar-fs, albeit using transductive inference (that is utilizing the query set as a set of additional unlabeled samples from the support classes) thus making a somewhat unfair comparison when put together with the inductive methods in the same table.

**Questions:**

please see weaknesses

**Limitations:**

not discussed

**Strengths And Weaknesses:**

Strength:
- I like the use of dense matching, cross attention and position (crop) and rotation refinement, I can see how those are useful for the set task
- good results were obtained, albeit in a transductive setup

Weaknesses:
- a proper comparison to transductive sota needs to be made, this sota can easily be found e,g, on papers-with-code:
https://paperswithcode.com/sota/few-shot-image-classification-on-mini-2
as we can learn from the numbers there the true transductive sota is much higher than what is compared in the current paper. The situation is similar also with the other benchmarks (tiered, cub, cifar-fs, etc)
- novelty (w.r.t. the RENet, CAN and other dense matching methods) seems to be limited, the added modules are basically attention-driven affine transformation modules that afaik have been applied before to this task
- writing could be improved, as it is written the method is not reproducible and the code is stated to be "proprietary" so no code is provided nor code release is planned - all this reduces the value of the current paper as no1 will be able to reproduce these results based on the description.

---

### Official Review · Reviewer_qcLF · 2022-07-11

**Rating:** 4
**Confidence:** 5
**Soundness:** 2 fair
**Presentation:** 4 excellent
**Contribution:** 3 good

**Summary:**

The paper proposes two cross-attention architectures that augment RENet to improve few-shot classification accuracy. Motivated by the increased performance of methods where attention is deployed effectively, CSA (cross-scaled attention) and CAA (cross-aligned attention) introduce a further hierarchy of attention to RENet to improve performance. The former improves feature matching by better scaling the maps while the latter mitigates the alignment issue by PCA. Extensive experiments demonstrate state-of-the-art performance while ablation studies justify the efficacy of each component.

**Questions:**

Please answer the questions noted in the above section with respect to the results. Specifically, first how much of the improved performance is simply a factor of the proposed architectures having more parameters than RENet. Second, following on that, do CSA and CAA improve performance over other simpler attention architectures that add the same number of parameters to RENet?

**Limitations:**

There is no explicit discussion of limitations or potential negative social impact of the work. The submission could benefit from a dedicated section that addresses the technical limitations of the work, lays out areas for future improvement, and explores potential societal impacts of the work.

**Strengths And Weaknesses:**

Strengths:
- Paper is very well-written and easy to follow (but some small typos in the related work section)
- Experiments and ablation studies demonstrate state-of-the-art performance of both CSA and CAA
- The motivation behind the proposed architecture is sound

Weaknesses:
- Details on parameter counts of RENet+CSA and +CAA are required to determine if the speed of inference reported is simply correlated with the parameter count of the model.
- In some tables, only the best values are in bold indicating state-of-the-art. However, when 95% confidence intervals are considered CSA and CAA are matching statistically speaking, and therefore, the claim for CSA's superior performance is much more marginal. The results classification and the resulting claims must be updated to reflect this.
- There are also two major questions from the results that weaken some of the claimed contributions: First, how much of the improved performance is simply a factor of the proposed architectures having more parameters than RENet. Second, following on that, do CSA and CAA improve performance over other simpler attention architectures that add the same number of parameters to RENet? This could be simply addressed by providing parameter count information and if possible, evaluating a simple standard attention architecture with a similar parameter count added to RENet.
- It is also noted that the architectures are evaluated in a transductive manner rather than inductively. Transductive evaluation gives a natural advantage in performance. Therefore, all results tables must be updated to indicate whether the baseline is transductive or inductive. Furthermore, inductive evaluation of CSA and CAA would be useful to see if there performance gain is directly from the architectures and not from the advantages in transductive inference.
- Paper is written around RENet and assumes considerable existing knowledge of the RENet architecture. This results in the language reading like a simple extension paper and some details being confusing before reviewing RENet.

---

> ### Comment · Reviewer_qcLF · 2022-08-09
> **Have the authors submitted a rebuttal?**
>
> I can't quite tell if there has been a new updated draft of the paper. If the authors have not submitted a rebuttal, without new information, I would continue to maintain my current recommendation for rejection.

---

### Official Review · Reviewer_bTff · 2022-07-12

**Rating:** 4
**Confidence:** 3
**Soundness:** 3 good
**Presentation:** 2 fair
**Contribution:** 2 fair

**Summary:**

This paper extends relational embedding network [18] by introducing two novel cross attention modules namely the cross scaled attention (CSA) and the cross aligned attention (CAA). While CSA normalizes different feature maps to be in the same scale, the CAA aligns features from different images via PCA. The experiments are conducted on four public datasets and show new state of the art results.

**Questions:**

The presentation of the results in Table 1 can be improved by grouping different methods into inductive, semi-supervised, transductive learning. Which compared methods in Table 1 belong to transductive learning?

**Limitations:**

I cannot find the limitations and potential negative societal impact in the paper.

**Strengths And Weaknesses:**

Strengths

1. The proposed cross-attention modules seem to be novel in few-shot learning field.
2. The experiments show significant improvement over the baselines, which looks quite promising

Weakness

1. Although the results in Table 1 are impressive, it is unclear if the comparison is completely fair. Specifically, this paper is in the transductive learning setting, which directly uses the test samples as the unlabeled samples, while many methods in Table 1 do not use any unlabeled examples or they are in the standard semi-supervised setting where the unlabeled samples are not the test samples. Comparing with inductive methods or semi-supervised methods with the proposed transductive method is not fair.

---

### Meta-Review · Area_Chair_JZH8 · 2022-08-26

**Recommendation:** Reject
**Confidence:** Certain

**Metareview:**

This paper studies the few-shot classification with the focus of extending RENet with cross scaled attention and cross aligned attention. The technical novelty is marginal, and the authors fail to provide feedback during the rebuttal. I recommend rejection.

**Award:**

No

---

### Decision · Program_Chairs · 2022-09-14

Reject